# Random Telegraph Noise Degradation Caused by Hot Carrier Injection in a 0.8 μm-Pitch 8.3Mpixel Stacked CMOS Image Sensor [note 1]

**DOI:** 10.3390/s23187959

**Published:** 2023-09-18

**Authors:** Calvin Yi-Ping Chao, Thomas Meng-Hsiu Wu, Shang-Fu Yeh, Chih-Lin Lee, Honyih Tu, Joey Chiao-Yi Huang, Chin-Hao Chang

**Affiliations:** Taiwan Semiconductor Manufacturing Company (TSMC), Hsinchu 30077, Taiwan; mhwuzd@tsmc.com (T.M.-H.W.); sfyehe@tsmc.com (S.-F.Y.); clleeza@tsmc.com (C.-L.L.); hytu@tsmc.com (H.T.); joey_huang@tsmc.com (J.C.-Y.H.); cchangr@tsmc.com (C.-H.C.)

**Keywords:** CMOS image sensor (CIS), random telegraph signal (RTS), random telegraph noise (RTN), MOSFET Channel RTN (MC-RTN), flicker noise, 1/f noise, hot carrier injection (HCI), hot carrier stress (HCS), hot carrier aging (HCA)

## Abstract

In this work, the degradation of the random telegraph noise (RTN) and the threshold voltage ($V_{t}$) shift of an 8.3Mpixel stacked CMOS image sensor (CIS) under hot carrier injection (HCI) stress are investigated. We report for the first time the significant statistical differences between these two device aging phenomena. The $V_{t}$ shift is relatively uniform among all the devices and gradually evolves over time. By contrast, the RTN degradation is evidently abrupt and random in nature and only happens to a small percentage of devices. The generation of new RTN traps by HCI during times of stress is demonstrated both statistically and on the individual device level. An improved method is developed to identify RTN devices with degenerate amplitude histograms.

## 1. Introduction

In the past 5 years, the pixel pitch of CMOS image sensors (CIS) has shrunk from 0.9 μm to 0.56 μm [1,2,3,4,5,6,7,8,9,10,11,12,13,14,15,16] at a fast pace of almost one generation every year. Despite the pixel design and process development effort to maintain the quantum efficiency (QE) and the full well capacity (FWC) at comparable levels across the generations, the smaller pixels still face the fundamental physical challenge: under a given scene illumination and integration time, the number of photons that can be captured by smaller pixels is inevitably less. Thus, to achieve a good signal-to-noise ratio (SNR), the sensor readout random noise (RN) becomes an increasingly important performance-limiting factor. In general, the known RN consists of the thermal noise, the flicker (1/f) noise, and the random telegraph noise (RTN). Although a median readout noise of 1 e-rms or less can be achieved by careful circuit design, the RTN on the noise distribution tail, typically dominated by the pixel source followers, could be more than 10 times higher than the median or average noise [1,2,3,10,17,18,19,20,21,22]. Therefore, understanding and reducing RTN are especially critical to achieving high dynamic range and good image quality for CIS products.

On the other hand, the hot carrier injection (HCI), together with the time dependent dielectric breakdown (TDDB), the bias temperature instability (BTI), and the electron migration (EM), are the most important aging and reliability issues for advanced CMOS devices [23,24,25,26,27,28,29]. Although the RTN degradation of MOS transistors due to HCI was known in the literature [30,31,32,33,34], the statistical characterization involving a large sample size was not reported before. In this work, we study the effects of HCI stress on RTN and the threshold voltage shift of NMOS in an 8.3Mpixel CIS. The test chip design and the experimental methods are described in Section 2. The results are presented in Section 3. The discussions and conclusions are given in Section 4 and Section 5, respectively.

## 2. Materials and Methods

### 2.1. Test Chip Architecture and Characteristics

The test chip is a stacked backside-illuminated (BSI) CIS with a pixel array on the top layer, fabricated by a 28 nm 1P4M CIS process, and a readout-circuit layer on the bottom by a 22 nm 1P7M mixed-mode process, stacked by a wafer-level hybrid bond (HB) technology. The array consists of 2512 × 3296 (8.3M) pixels with a 0.8 μm pitch and a 4 × 2-shared structure, read out by a bank of 1648 column-parallel 12-bit ADCs with front-end amplifiers supporting 1× to 8× analog gains. A simplified analog signal-chain schematic is shown in Figure 1. The three pixel devices, the reset (RST), the row select (RSL), and the source follower (SF), support 3.3 V operation. The device under study is the SF NMOS with W = 0.16 μm, L = 0.87 μm, and a 5.7 nm dielectric thickness, biased by a constant column current source (CB) of 7.2 μA in normal operations. The conversion factor from the ADC output to the SF output is 292 μV/DN at 1× gain and 36.5 μV/DN at 8× gain. The median read noise is about 190 μV-rms at 8× gain, operated at a 60 MHz clock and a 1.48 fps frame rate. The circuit-only noise is about 115 μV-rms measured in a test mode with external reference voltages.

### 2.2. Hot Carrier Injection Characterization and Hot Carrier Stress Experiments

In normal imaging mode, the SF output ($V_{O}$) is read out by correlated double sampling (CDS), before and after the charge transfer from the photodiode to the sense node (SN). The voltage difference is amplified and digitized by column ADCs. In this work, we are only concerned with the SF device. The charge transfer is disabled, and the SF gate voltage ($V_{G}$) is fixed by RSV with RST turned on. The readout random noise (RN) is measured with CDS at 8× gain. The $V_{GS}$ of SF is obtained in a test mode by measuring the difference between $V_{O}$ and a fixed external reference voltage at 1× gain without CDS.

The SF drain voltage ($V_{D}$), or the $V_{PIX}$ in Figure 1, is 3.1 V for normal operation. When $V_{D}$ is increased, the channel conduction electrons are accelerated by the high electric field in the space-charge region determined by $V_{DS}-V_{Dsat}$, where $V_{Dsat}\approx V_{GS}-V_{t}$ is the minimal $V_{DS}$ for the transistor to operate in the saturation condition. The energetic and hot electrons can in turn generate more electron-hole pairs through the process of impact ionization. This is characterized by the rapid increase of the substrate current ($I_{B}$) due to the holes as $V_{D}$ increases. Figure 2a shows the measured $I_{B}$ of a SF device versus $V_{D}$ under various gate voltages ($V_{G}$) in a separate test key on the same wafer as the CIS chip. The body of the device is grounded to 0 V and the source is biased by a constant current source of 8 μA to mimic the pixel operation. As $V_{G}$ increases, the source voltage $V_{S}$ increases accordingly and the threshold voltage $V_{t}$ is also increased due to the body effect.

Figure 2b shows that all the data points in Figure 2a fall onto a universal curve when $I_{B}$ is plotted against $V_{DS}-V_{Dsat}\approx V_{D}-V_{G}+V_{t}$ with $V_{t}$ as an empirical fitting parameter. It also shows that $I_{B}$ increases rapidly when $V_{DS}-V_{Dsat}$ exceeds 1.5 V, since the hot electrons need at least 1.1 eV kinetic energy to start impact ionization.

Furthermore, Figure 2c shows that the substrate current due to the impact ionization can be described by a well-known formula [35,36,37]:(1)$$I_{B}=P_{1}I_{S}\left(V_{DS}-V_{Dsat}\right)\cdot exp\left(-P_{2}/\left(V_{DS}-V_{Dsat}\right)\right),$$
where $P=\left(P_{1},P_{2}\right)$ are two fitting parameters and $I_{S}$ is the source current. It is clear that the measured data follow Equation (1) very well over more than six decades of the current range. We also measured the reverse-biased drain-to-body junction leakage, and it was in the range of 1 to 10 pA, negligible compared to $I_{B}$ when $V_{DS}-V_{Dsat}>$ 2 V.

In typical reliability testing, the supply voltage is raised to 20–50% higher than its normal value to speed up the device’s aging. In our experiments, the SF is biased by a 7.2 μA source current ($I_{S}$), and we choose to raise the $V_{D}\overset{\mathrm{def}}{=}V_{PIX}$ in Figure 1 up to 6.6 V where the $I_{B}$ becomes comparable to $I_{S}$ according to Figure 2a. Several devices under test (DUTs) were selected and stressed for 10, 20, 50, and 100 min. Between two consecutive stresses, the DUT is measured under normal operation conditions. Negligible relaxation effects were observed for several days after stressing. All experiments and measurements were performed at room temperature.

Because of the row-by-row rolling readout operation and the fact that there are 628 rows of SFs in the pixel array, the stress imposed on the DUT is different when it is active (RSL turned on, about 1/628 of the nominal stress time) and when it is deactivated (RSL turned off, about 627/628 of the nominal stress time). Therefore, the actual time the DUT is under HCI stress is much smaller than the nominal stress time. It requires further study to distinguish these two stress effects and their dependencies on stress voltages.

## 3. Results

Figure 3 shows the SF bias configuration during the stress experiments and illustrates the physical mechanism of the hot carrier degradation. As $V_{D}$ increases, the channel conduction electrons are accelerated by the high electric field in the space charge region. The energetic hot electrons can generate more electron-hole pairs through the process of impact ionization near the drain end. Some of these electrons and holes will recombine and release their energies into the surrounding Si lattice. Some electrons and holes could recombine to emit photons even though Si is known as an indirect-bandgap material. In fact, it was reported in [38] that the photon emission caused by the hot carriers from the circuits in the bottom layer of a stacked CIS can be readily detected by the pixel array in the top layer. The rest of the unrecombined electrons and holes are separated by the electric field in the depletion region near the drain. The holes going to the substrate contribute to the body currents. The hot electrons may overcome the energy barrier (about 3.2 eV) at the Si-SiO_2_ interface, or through the direct or indirect tunneling process, and get injected into the gate oxide to cause damage and generate traps.

### 3.1. Threshold Voltage Shift and Random Noise Degradation

Two well-known device aging effects caused by HCI are threshold voltage shift and mobility degradation [25]. In our SF configuration, either the increase of $V_{t}$ or the decrease of mobility lead to a similar decrease of the SF output voltage. Since the $V_{G}$ and $I_{S}$ of the SF are fixed in our setup, the change in SF output is considered an effective $V_{t}$ shift ($\Delta V_{t}$) in the rest of this paper. Figure 4 shows the measured distribution of $V_{GS}$ after a series of HCI stresses up to 100 min, with $V_{G}$ = 2.8 V, $V_{D}$ = 6.6 V, and $I_{S}$ = 7.2 μA.

The change of $V_{GS}$ (effective $\Delta V_{t}$) is small and not much noticeable in Figure 4. However, the $\Delta V_{t}$ becomes very clear in Figure 5a histograms by subtracting the unstressed $V_{GS}$ from the stressed $V_{GS}$ for each stress time t, $\Delta V_{t}\left(t\right)\overset{\mathrm{def}}{=}\Delta V_{GS}\left(t\right)-V_{GS}\left(0\right)$. The corresponding inverse cumulative distribution function (ICDF) family of curves is plotted in Figure 5b. The systematic increasing trend of $\Delta V_{t}$ of various constant-ICDF contours is shown in Figure 5c.

The random noise (RN) is calculated as the standard deviation of 100 consecutive frames of data, device by device. The change of RN before and after HCI stress is calculated as $\Delta RN\left(t\right)\overset{\mathrm{def}}{=}\sqrt{{\left(RN\left(t\right)\right)}^{2}-{\left(RN\left(0\right)\right)}^{2}}$. The histograms, the ICDF curves, and the constant ICDF contours of ∆RN after 10-, 20-, 50-, and 100-min stress are shown in Figure 6a–c, respectively.

Apparently, there are similarities between Figure 5a–c and Figure 6a–c. Both histograms (or PDFs, the probability density functions) and ICDF curves show skewed non-Gaussian distributions with noticeable long tails. The degradation of the entire population seems systematic and uniform at every chosen ICDF level from the median (ICDF = 0.5) to 5 ppm and is approximately linearly dependent on the logarithm of the stress time. However, the similarities are only superficial. A close examination of the details in the next section shows there are significant differences as well.

### 3.2. Key Findings: The Differences between Vt Shift and RN Degradation

The differences in the degradation of $V_{t}$ and RN become clear when we look at how the individual devices change progressively throughout the stress time. Figure 7a–c show the correlation scatter plots of the $\Delta V_{t}$ after 20-, 50-, and 100-min stress versus that after 10-min stress. Regardless of the device-to-device variation, we observe that all the 1M devices degrade quite uniformly. All the data points are concentrated in the neighborhood along a straight line in each plot. The red dash lines are the best-fit results. The linear fit x/y ratios of 1.78, 2.56, and 3.35 shown in Figure 7a–c are roughly proportional to the ratios of the logarithm of the stress time.

By contrast, the correlation of the RN after 10-, 20-, and 100-min stress versus that before stress plotted in Figure 8a–c shows quite distinctive features. The scatter plots reveal two obvious branches. In each graph, one group of data points centers along the x = y diagonal line, representing the devices whose RNs remain almost unchanged after stress. The other group of data points on the lower-right branches correspond to devices whose RN significantly increased after longer stress times. It is also clear that the number of data points in the lower-right branch continues to increase as the stress time increases. A closer examination of the noises of the devices in the lower-right branch shows that they are dominated by RTN, as illustrated in Section 3.3.

The difference between the $V_{t}$ shift and the RN degradation can be further highlighted in a side-by-side comparison of the 2D histograms of the correlation in Figure 9.

### 3.3. Tracking the Degradation of Individual Devices

Moreover, we could track and monitor the individual devices to see how they evolve over time under stress. Figure 10a–c show the snapshots of the signal waveforms and their corresponding amplitude histograms of 5000 samples for three selected devices after 0-, 20-, and 100-min stress, from top to bottom. The (row and column) coordinates of the cells are given in the plots. The device in Figure 10a shows a single RTN trap before the stress and remains unchanged after 20- and 100-min stress. The feature of two side peaks equally spaced on both sides of a center main peak is the consequence of correlated double sampling (CDS) [17,18,19,20,21,22]. The device in Figure 10b shows no trap before stress; however, one single-trap defect is apparently generated after 20 min of stress and stays the same up to 100 min of stress. Figure 10c shows a device with no RTN trap after 20 min of stress, and one RTN trap is generated after 100 min of stress. These are concrete examples of trap generation during HCI stress. Since the RTN trap generation is a random and abrupt process, this explains why the RN scatter plots in Figure 8 split into two distinct branches.

A defect-related RTN trap may be caused by process-induced damage (PID) or by HCI stress. Once a defect is generated, it is not expected to disappear through further stress. However, Figure 11a–c show interesting examples where the three discrete RTN peaks appear to become degenerate and merged into 1 peak due to the broadening of the peak width. We speculate that it is caused by the increase of non-RTN noises such as the flicker noises due to the degradation of interface qualities.

Figure 11 suggests that a waveform and amplitude histogram showing detectable discrete levels may be a sufficient but not necessary condition to distinguish a RTN device from a non-RTN device. This observation motivates us to improve the algorithm used in [17,18,19,20,21,22] to sort out and count the number of RTN devices automatically for large-size samples, to be discussed in the next section.

### 3.4. Idenfying the RTN Devices

To quantify the RTN generation during the HCI stress, we like to count the numbers of RTN devices among the 1M cells and show that the number indeed increases as the stress time increases. The signature of RTN behavior is the clearly observable discrete states in time-domain waveforms, as exemplified in Figure 10. Previously [17,18,19,20,21,22], we developed a peak-finding program in MATLAB to search for the local maxima in the amplitude histograms. If more than one peak is found, we designate the device as an RTN device. Such a method is not precise for several practical reasons. For instance, if the histograms are not sufficiently smoothed, some data glitches could be misidentified as RTN peaks. Or, if the RTN amplitude is small or the non-RTN noises are high, the histograms do not show identifiable discrete peaks, such as in the degenerate cases in Figure 11. To deal with the degenerate cases, we developed an enhanced algorithm illustrated in Figure 12, where we fit the center peak of the histogram (the black curve, h[n]) with an ideal Gaussian distribution (the blue curve, g[n]) and calculate the deviation as the ratio *R* of the area under the red curve (|h[n]−g[n]|) versus the whole area under the black curve:(2)$$R\overset{\mathrm{def}}{=}\left(\sum_{n}\left|h\left[n\right]-g\left[n\right]\right|\right)/\left(\sum_{n}h\left[n\right]\right),\;\mathrm{where}\;n\;\mathrm{is}\;\mathrm{the}\;\mathrm{bin}\;\mathrm{index}.$$

In essence, the ratio *R* is a measure of how much the real histogram deviates from the ideal Gaussian shape. For a perfect Gaussian distribution, *R* equals zero. If *R* is larger than a certain threshold (typically 15~25%), we consider the histogram to be sufficiently different from a Gaussian distribution and designate the device as an RTN-like device. This decomposition method helps to reveal the red-area side peaks, which are otherwise hidden and indistinguishable by inspecting the waveforms.

The above criterion is based on the implicit assumption that the amplitude distribution of thermal and flicker noises is close to Gaussian [39]; therefore, any amplitude distributions sufficiently deviant from Gaussian are RTN-like. In fact, we have verified that for the majority of non-RTN devices, the amplitude distributions are indeed close to Gaussian. Two examples are given in Figure 13a,b, with ratio *R* equal to 7% and 9%, respectively.

With the methodology described above, we can sort all 1M devices into two groups, RTN (including RTN-like) and non-RTN, and show the changes over stress time in Figure 14. In the legends, N3 represents the number of devices showing three or more distinct peaks in the amplitude histograms, and N2 is the number of devices showing two or more identifiable peaks. These are RTN devices, without doubt. The Nx is the sum of N2 and the number of RTN-like devices such as those in Figure 12, with the empirical *R*-threshold set to 15%. The blue curve is the histogram of the remaining non-RTN devices, and its number is labeled as N1. Figure 14 shows that the number of RTN devices (Nx) is slightly less than 1% of the total before stress and gradually increases to about 4% after 100 min stress. It is also clear that the non-RTN noise tails (thermal and flicker noises) are increased by the HCI stress as well.

Finally, the RTN counts for all stress times are summarized in Figure 15. Since there is no well-defined boundary separating the RTN and non-RTN devices, the sorting and counting results are understandably approximations. When the *R*-threshold is set to a smaller value, more RTN-like devices will be counted; vice versa if the *R*-threshold is set to a larger value. To estimate the uncertainty of the sorting method, the results of setting different empirical thresholds of *R* = 10%, 15%, and 20% are compared in Figure 15. Although the identifiable RTN numbers depend on the choice of an arbitrary threshold, the increasing trend versus stress time is clear and consistent. The RTN and RTN-like device counts increase from 0.8~1.3% before stress to 3~6% after 100-min stress. This sorting allows us to quantify the RTN degradation.

### 3.5. Voltage Dependency

The data discussed prior to this section were overstressed under $V_{D}$ = 6.6 V and a stress time of up to 100 min. We observed that the degradation was strongly dependent on the stress voltage. In the following, we present more results under various lower voltages and longer stress times, up to 1600 min. The stress conditions are listed in Table 1 below, where “Y” means included and “N” means not included in the experiments.

The overall $V_{t}$ shift and RN degradation are plotted in Figure 16a,b against the effective stress defined below:(3)$$\mathrm{Effective}\mathrm{stress}\overset{\mathrm{def}}{=}t\cdot I_{B}\left(V_{D}\right)\cdot F_{EFF}(V_{D})\;(\mathrm{unit}:\;\mu A*\min),$$
where t is the nominal stress time, $I_{B}\left(V_{D}\right)$ is the substrate current, $V_{D}$ is the stress voltage, and $F_{EFF}\left(V_{D}\right)$ is an empirical voltage-dependent effectiveness factor. The $V_{G}$, $V_{B}$, and $I_{S}$ of the DUT are fixed as described in Section 2. The number of hot carriers created in the stress experiment is considered proportional to the product of substrate current ($I_{B}$) and stress time (t). In addition to the substrate current, the effect of stress should also strongly depend on the average kinetic energy of the hot carriers, which is a function of the stress voltage. Neither the energy distribution of hot carriers nor their effectiveness in causing device damage are directly measurable. Therefore, an empirical effectiveness factor $F_{EFF}\left(V_{D}\right)$ is tentatively introduced in Equation (3) to define the effective stress. This is similar to the acceleration factor defined as the ratio of the time-of-failure under normal operation conditions versus that under stress conditions in reliability modeling [27] (pp. 137–148).

In Table 1, the effectiveness factor is arbitrarily set to 1 for $V_{D}$= 5.5 V as a reference point and the values for other voltages are obtained by fitting the data such that the constant-ICDF contours for all voltages could be pieced together into a smooth and continuous family of curves as shown in Figure 16. The data in Figure 16a are fit by straight lines, and the data in Figure 16b by piecewise second-order polynomials. For example, Table 1 suggests that stressing under $V_{D}$= 6.6 V is about 100 times more effective in causing device damage than stressing under $V_{D}$= 5.5 V. The same factors are used in plotting the $V_{t}$ shift in Figure 16a and the RN degradation in Figure 16b. The level-off of the RN degradation in the lower effective stress region is probably an artifact due to the uncertainties of the RN calculation by RMS in limited samples. The monotonic increasing trends are clear and consistent in Figure 16a,b. Further studies are needed to understand the physical meaning and justify the use of empirical factors. It is beyond the scope of this work to establish a universal aging model that may be used to extrapolate the data to lower stress voltages but a much longer lifetime, e.g., 5~10 years typically required for commercial products.

## 4. Discussion

In many test chips and technology nodes we have studied so far [17,18,19,20,21,22], the percentage of RTN devices is small, typically in the order of 1%, after wafer-out and without any external stress or deliberate radiation damage. In other words, all the MOSFET devices have thermal and flicker noises; however, only a small percentage of them show RTN behaviors. Furthermore, the dominant RTN type observed is always the single-trap RTN. Multiple-trap RTN devices do exist but are relatively rare. This may be understood as a simple statistical argument. Assuming the defects causing RTN are random and independent events, if the probability of having one defect is p≪1, then the probability of having two defects in the same device is roughly $p^{2}\ll p$. It is also known that p depends strongly on device types, sizes, bias conditions, and detailed process recipes. The small RTN percentage implies that it is important to measure a large number of samples to fully characterize the RTN behavior or to draw firm conclusions from any process improvement experiments.

The boundary between RTN and flicker noise is blurry and probably cannot be well defined. When the RTN amplitude is comparable to the flicker/thermal noise, it becomes difficult to distinguish one from the other. The approach adopted in this work to identify and count RTN devices has some limitations and uncertainties, as illustrated in Figure 15. Perhaps better methods will be developed in the future. Another way to separate the RTN from flicker noise is to examine the noise power spectral density (PSD) in the frequency domain. The RTN spectra is a Lorentzian-shape with a $1/f^{2}$ roll-off [40] while the flicker noise shows a 1/f slope up to a corner frequency where the thermal (white) noise becomes dominant. However, the noise spectra measurement on a large array has not been reported so far. The statistical correlation between the time-domain waveforms and the frequency-domain spectra is another area worth further investigation.

## 5. Conclusions

In this paper, we report the RTN degradation caused by HCI stress in a CIS chip both statistically and at the level of individual devices. The key finding highlighted the characteristic differences between the effective $V_{t}$ shift and the RTN degradation, although their physical mechanisms might be similarly related to the generation of interface states and traps inside the gate dielectric by hot carriers. The $V_{t}$ shift is more gradual and uniform among all devices. RTN degradation is a random and abrupt change that happens only in a small percentage of devices. In the measure-stress-measure experiments, we can locate the exact time interval when a new RTN trap is generated in any specific device. The observation is a general device aging phenomenon that may have negative impacts on a variety of circuits and products other than CIS. Monitoring the RTN degradation could be a useful tool to study the long-term reliability of MOSFET devices. A more systematic study on a wider range of voltages, stress times, and temperatures is needed.

## Figures and Tables

**Figure 1 sensors-23-07959-f001:**
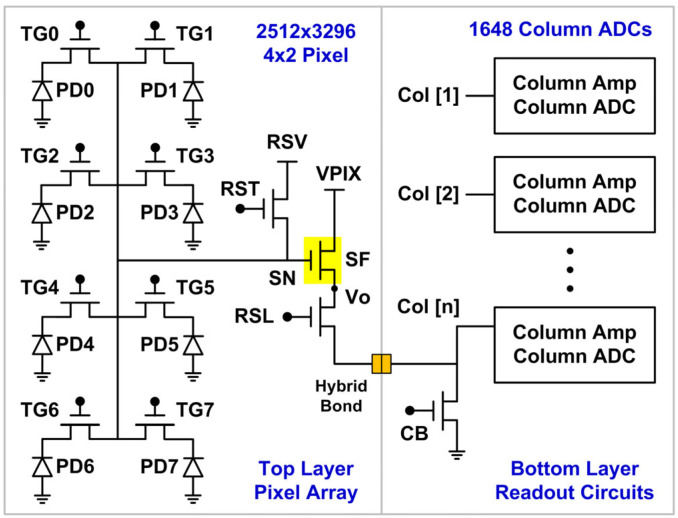
Simplified test chip architecture. The device under stress is the source follower (SF) NMOS in the 4 × 2-shared pixels on the top layer. The PD0–7 are the photodiodes, and the TG0–7 are the transfer gates in each 4 × 2-shared pixel. The total number of SF is 628 × 1648 = 1.03 M.

**Figure 2 sensors-23-07959-f002:**
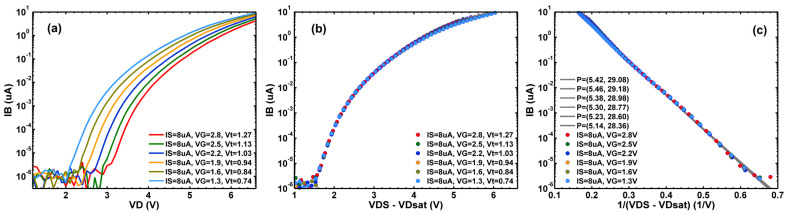
(**a**) The measured $I_{B}$ of a SF device vs. $V_{D}$ with $V_{G}$ stepping from 1.3 V to 2.8 V; (**b**) The same data as in (**a**) but plotted against $V_{DS}-V_{Dsat}\approx V_{D}-V_{G}+V_{t}$ with $V_{t}$ as a fitting parameter; (**c**) The same data as in (**b**) plotted against $1/\left(V_{DS}-V_{Dsat}\right)$ with $P=\left(P_{1},P_{2}\right)$ as two fitting parameters according to Equation (1).

**Figure 3 sensors-23-07959-f003:**
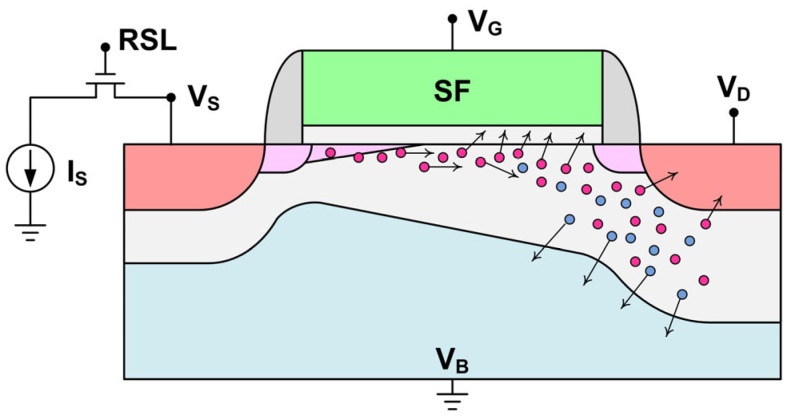
The bias configuration of the SF under test. The red and blue solid circles symbolize electrons and holes, respectively.

**Figure 4 sensors-23-07959-f004:**
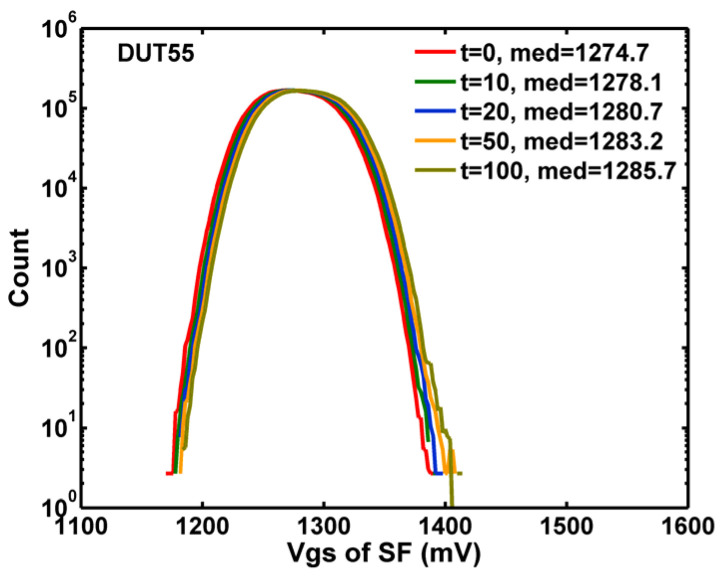
The histograms of the measured $V_{GS}$ of the SF for stress time (t) from 0 to 100 min.

**Figure 5 sensors-23-07959-f005:**
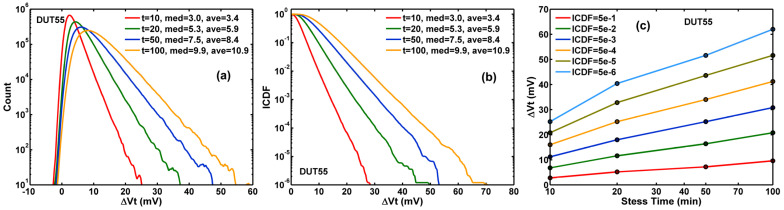
(**a**) The histograms of the threshold voltage shift ($\Delta V_{t}$) after 10-, 20-, 50-, and 100-min stress; (**b**) The inverse cumulative distribution function (ICDF) curves of $\Delta V_{t}$; (**c**) the constant ICDF contours against stress time (t).

**Figure 6 sensors-23-07959-f006:**
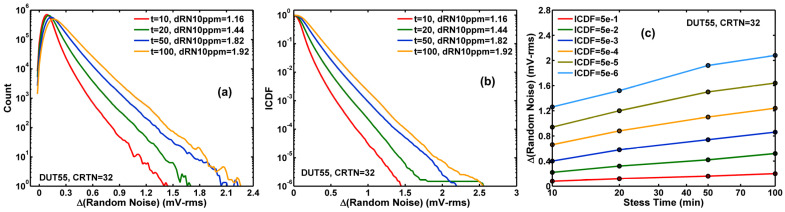
(**a**) The histograms of the random noise changes (ΔRN) after 10, 20, 50, 100 min stress; (**b**) The inverse cumulative distribution function (ICDF) curves; (**c**) the constant ICDF contours as functions of stress time (t).

**Figure 7 sensors-23-07959-f007:**
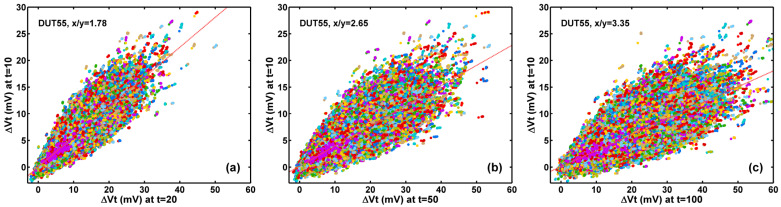
The correlation of the SF threshold voltage shift ($\Delta V_{t}$) after 10 min of HCI stress vs. after (**a**) 20 min, (**b**) 50 min, and (**c**) 100 min of stress, respectively. The linear least-square fit of the x/y ratio (red dash line) shows the continuous increase of the $\Delta V_{t}$ as the stress time increases. The $\Delta V_{t}$ increases are relatively uniform among all 1M devices, which is quite different from the random noise increases in Figure 8 below. Random colors are assigned to the data points to separate the dots from each other.

**Figure 8 sensors-23-07959-f008:**
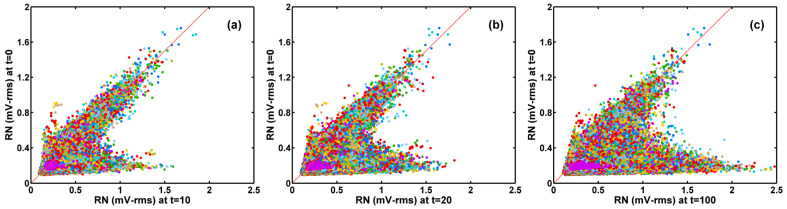
The correlation of the random noises (RN) before HCI stress (t = 0) vs. after (**a**) 10 min, (**b**) 20 min, and (**c**) 100 min stress, respectively The RN increases are noticeably nonuniform. The RN along the x = y red dash line remains relatively unchanged. The devices on the lower-right branches show a significant increase in RN. The population of the lower branch increases as stress time increases. Random colors are assigned to the data points to separate the dots from each other.

**Figure 9 sensors-23-07959-f009:**
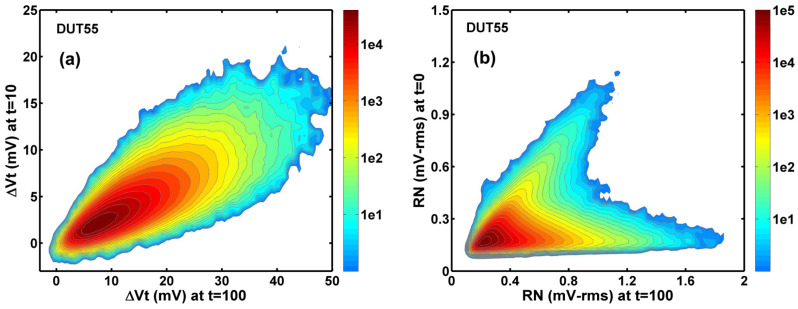
The 2D histograms of the correlation of the $V_{t}$ shift and RN degradation shows dramatically different statistical behaviors. (**a**) The $V_{t}$ change after 100-min stress versus that after 10-min stress. (**b**) The RN after 100 min stress versus that before the stress.

**Figure 10 sensors-23-07959-f010:**
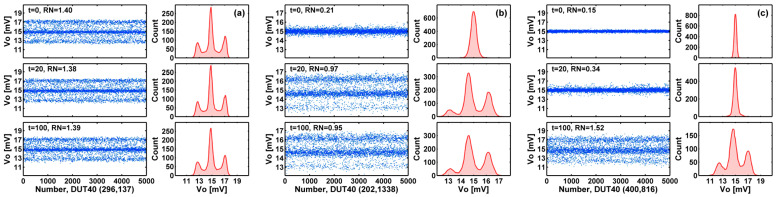
Generation of RTN traps during HCI stress. The 5000-frame waveforms before (t = 0) and after the HCI stress (t = 20, 100 min) with the corresponding histograms are shown for three selected examples. (**a**) Device (296, 137) shows one trap before stress and remains unchanged after stress. (**b**) Device (202, 1338) shows no trap before stress and one trap generated after 20 min of stress. (**c**) Device (400, 816) shows no trap before stress; however, one trap is generated after 100 min of stress. The RN unit is mV-rms.

**Figure 11 sensors-23-07959-f011:**
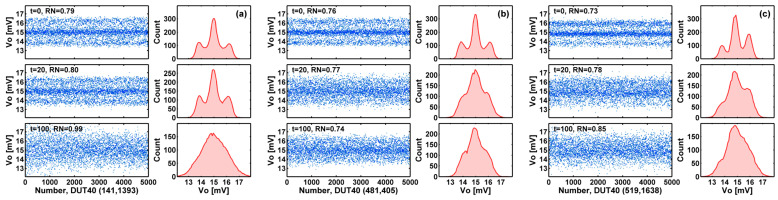
Degeneration of the RTN discrete levels. During HCI stress, the non-RTN noises may be increased significantly such that the discrete RTN levels become indistinguishable. (**a**) Device (141, 1393) show such degeneration after 100 min of stress. (**b**) Device (481, 405) show degeneration after 20 min of stress. (**c**) Device (519, 1638) shows unsymmetric side peaks and unsymmetric degeneration after 20 min and 100 min of stresses. The RN unit is mV-rms.

**Figure 12 sensors-23-07959-f012:**
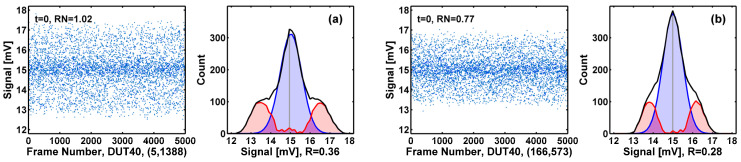
For devices showing a single histogram peak, if the histogram is significantly different from the Gaussian distribution, they are counted as RTN-like devices. The ratio *R* expressed in Equation (2) is defined as the red area versus the total area under the black histogram. The *R* values in examples (**a**) and (**b**) are 36% and 28%, respectively. The RN unit is mV-rms.

**Figure 13 sensors-23-07959-f013:**
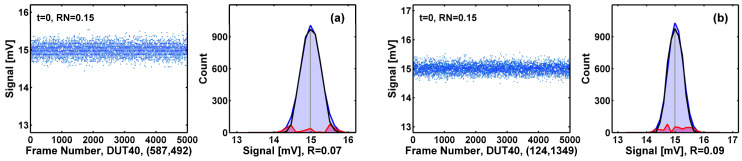
Devices with amplitude distributions close to Gaussian are considered as non-RTN devices. The deviation ratio *R* is 7% for device (587, 492) in (**a**) and 9% for device (124, 1349) in (**b**). The RN unit is mV-rms.

**Figure 14 sensors-23-07959-f014:**
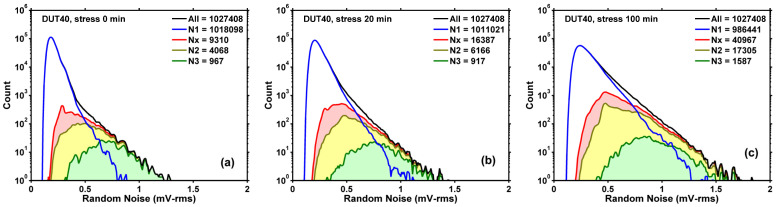
The RN distribution of the RTN and non-RTN devices, sorted by the improved algorithm: (**a**) before HCI stress, (**b**) after 20 min stress, and (**c**) after 100 min stress. The RTN devices clearly contribute to and dominate the long tails of the RN histograms. The number of RTN devices (Nx) (with the *R*-threshold set to 15%) increases systematically as the stress time increases.

**Figure 15 sensors-23-07959-f015:**
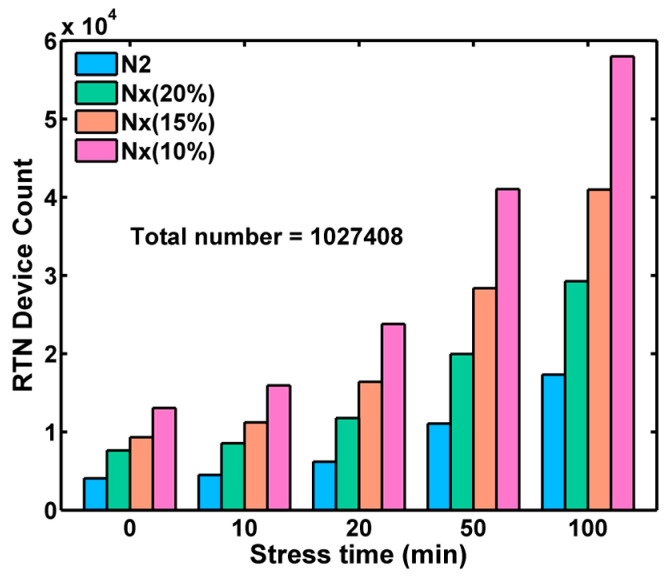
The count of RTN devices increases consistently as stress time increases. N2 is the number of devices showing two or more peaks in amplitude histograms. Nx is N2 plus the number of RTN-like devices determined by setting the *R*-threshold to 10%, 15%, and 20%, respectively.

**Figure 16 sensors-23-07959-f016:**
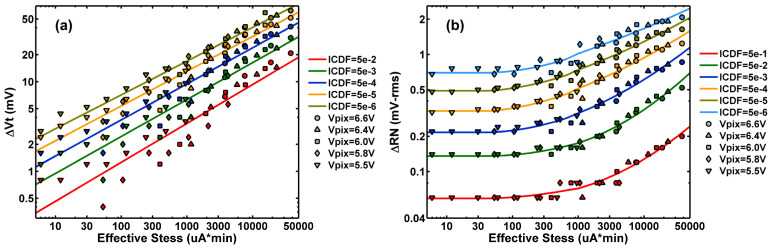
(**a**) The $V_{t}$ shift and (**b**) the RN degradation trends against the effective stress defined in Equation (3), where the effectiveness factors are treated as empirical fitting parameters such that all the constant-ICDF points for different voltages fall onto a family of continuous and smooth curves. The fitting results are listed in Table 1.

**Table 1 sensors-23-07959-t001:** The voltage and stress time combinations of the HCI experiments.

Stress Time (min)	0	10	20	50	100	200	400	800	1600	$I_{B}$(uA)	$F_{EFF}\left(V_{D}\right)$
$V_{D}=5.5\;V$	Y	Y	Y	Y	Y	Y	Y	Y	Y	0.597	1.0
$V_{D}=5.8\;V$	Y	Y	Y	Y	Y	Y	Y	Y	N	1.059	5.0
$V_{D}=6.0\;V$	Y	Y	Y	Y	Y	Y	Y	N	N	1.552	25
$V_{D}=6.4\;V$	Y	Y	Y	Y	Y	Y	N	N	N	2.897	40
$V_{D}=6.6\;V$	Y	Y	Y	Y	Y	N	N	N	N	3.789	100

## Data Availability

Not available.

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
