# Peer review of "Random Telegraph Noise Degradation Caused by Hot Carrier Injection in a 0.8 μm-Pitch 8.3Mpixel Stacked CMOS Image Sensorâ€"

_sensors, 2023, doi:10.3390/s23187959_

Round 1
Reviewer 1 Report
Please see the attached file.

Reviewer 2 Report
Figures are way too small, legends are not readable. Needs improvement.
Some sentences are extremely long. Shorten the sentences please.
Reviewer 3 Report
In this paper, the authors evaluate pixel degradation caused by hot carrier injection in a stacked CMOS image sensor based on actual measurements. Accelerated tests by applying stresses stronger than the actual operating conditions show a clear trend. In addition, evaluations of individual pixel circuits in actual arrayed devices show that there are differences in the distribution of threshold shift and random noise increase. These are considered to be sufficient contributions to the field. The quality of the paper is considered high enough, but here are some comments from the reviewer.
1. Some definitions are not clearly stated, such as RSV, IS, and t in Fig. 4.
2. “he” in line 294 seems to be a typographical error.
3. Please describe how the fitting curve in Fig. 16(b) was obtained.
